# Pathways to Sustainable Deployment of Solar Photovoltaic Policies in 20 Leading Countries Using a Qualitative Comparative Analysis

**Yujie Lu** [1,2,3], **Fangxin Yi** [4,*], **Shaocong Yu** [5], **Yangtian Feng** [5] **and Yujuan Wang** [6]

1   Department of Building Engineering, College of Civil Engineering, Tongji University, Shanghai 200092, China; lu6@tongji.edu.cn
2   Key Laboratory of Performance Evolution and Control for Engineering Structures of Ministry of Education, Tongji University, Shanghai 200092, China
3   Shanghai Institute of Intelligent Science and Technology, Tongji University, Shanghai 200092, China
4   Innovation Centre for Risk Governance, School of Social Development and Public Policy, Beijing Normal University, Beijing 100875, China
5   College of Management, Zhejiang University of Technology, Hangzhou 310023, China; yu.shaocong@outlook.com (S.Y.); tian.yt.feng@outlook.com (Y.F.)
6   Inner Knowing Coaching, Singapore 119077, Singapore; wangyjjuanrr@gmail.com
*   Correspondence: fangxin.yi@bnu.edu.cn

**Abstract:** The paper investigates the pathways and combinations of factors for the sustainable development of solar photovoltaic policies using a QCA analysis of 20 leading countries. The main finding of this research is the causal relationship between the selected contributing factors and sustainability of the policy outcomes, which is interpreted as high/low GDP with a high democracy level, high fossil-fuel consumption and high LCOE being related to the deployment of market-based policies which include target, FiT and others (subsidies, tax, loans, TGC/RPS); while high/low GDP, low level of PV penetration, high RE investment, and high R&D expenditure contributes to more successful technological-R&D-based policies which include R&D funding and demonstration programs.

**Keywords:** pathways; combinations of the factors; solar photovoltaic policies; QCA

## 1. Introduction

Solar electricity is now the fastest-growing energy source in the world and third most-important renewable energy source (RES) in terms of installed capacity worldwide [1]. The deployment of appropriate supportive policies has been the main driver of solar markets, as it makes an impact on the adoption of solar energy, the reduction in solar PV's electricity cost and the development of solar-related technologies [2,3]. In European countries, a retroactive subsidy change decreased the investment rate by approximately 45% for solar photovoltaics (PV) [1]. Particularly, the growing number of installations of PV was highly related to strong policy incentives in countries such as Germany, Spain and Italy [4]. Globally, various forms of policies have been deployed by different countries and some examples of the commonly adopted policies are feed-in tariffs, subsidies, tax reductions, and funded R&D of solar PV technology [5]. A literature review identifies that the support policies can be categorized into supply-side policies, economic development strategies and demand-side policies [6]; or, they can be categorized into demand-pull and technology-push policies [7], demand-side and supply-side policies, market incentives and technological and R&D incentives [8].

Critical factors that could potentially affect the deployment of PV support policies in different countries and contexts have also been identified [4]. For instance, among 18 Asian countries, those with a higher GDP per capita and democracy level tend to have more advanced policies launched, compared to poorer and authoritarian countries. In addition,

through a comparative study on China and India [9], it Was found that the amount of investment granted by the government towards the R&D of PV-related technologies, as well as current degrees of PV development and penetration, both have positive effects on the likelihood of the successful adoption of PV-related policies. In addition, in the context of America, the states with a stronger reliance on fossil fuels were generally less supportive of laws that promote renewable and clean-energy generation, such as using PV.

It should be noted that achieving the successful implementation of solar-PV policies requires an understanding of the conditions that enable the sustained functionality of combined factors. Most of the aforementioned literature either placed emphasis on a limited number of affecting factors in several countries, or intensively investigated various possible factors in the context of only one country or region, which has made research findings less applicable globally, as different countries have distinctive situations, especially for developed and developing countries.

Therefore, this research intends to identify the causal relationship between a combination of contributing factors and the various solar-PV policy outcomes towards sustainability in different contexts globally, through the QCA method. It uses fuzzy-set qualitative comparative analysis (fsQCA) to evaluate the combinations of causal conditions associated with sustainable policies' outcomes through a global set of 20 countries. It intends to identify which contributing factor (or combination of factors) is related to sustainable solar-PV market development through supportive policies.

## 2. Background and Literature Review

### 2.1. Global Solar-PV Policies Research and Categories of Policies

2.1.1. Global Solar-PV Policies Research Overview

According to an IEA report, 99% of the global PV market depends on support schemes or PPAs above the market price, and only 1% of the global PV has been driven by pure self-consumption, which has slightly increased compared to the 4% seen in 2014 due to regulatory driven policies. However, these support policies take various forms in different countries across the world. The main incentives adopted by countries in 2016 are: feed-in tariff (FiT) (59%), direct subsidies or tax breaks (22%), incentivized self-consumption or net-metering (10%), trading of green certificates (TGC) or similar RPS-based schemes (4%), feed-in tariff through tender (4%), and competitive PPA (1%) [10].

There is an abundance of research on a single case of a policy and multi-country policies at a national level. Some research focuses on a single country's policy with in-depth analysis, such as the strategic deployment of solar-PV policies in China [11], the evolution of Germany's FiT solar-PV policy [12], or the deployment of a FiT policy in Japan. Moreover, there is comparative research in two or more countries, such as a comparative analysis of Germany and China, and with Italy [13]; policies between Germany and Japan [14]; and the FiT incentive between Germany and Taiwan. In addition, there is some other research on solar policies at a regional level, such as research on the top-ten solar-PV-power-producing countries, or research on the solar policies in seven countries to advise Malaysia on developing strategies [5], or research on the development history of the US, German and Japan with a comparison to that of China, with advice on demand-side policies [15].

2.1.2. Categorization of the Global Solar-PV Policies

Due to the inconsistency of policies used in different countries, it will be difficult to compare the effectiveness of the various policies on a level ground, therefore making it difficult to investigate the relationship between contributing factors and policies. Thus, a literature review was performed to categorize policies into two main categories with a few representative policies under each of them. Table 1, below, shows different ways of categorization adopted by some of the recent studies on solar-PV policies. For example, support policies were categorized into supply-side policies, economic development strategies and demand-side policies [6]; or they were categorized into demand-pull and technology-push

policies [7], demand-side and supply-side policies [15], market incentives and technological and R&D incentives [16]).

**Table 1.** Literature review on solar-PV policy categorization.

| Authors/Papers | Categories | Detailed Policies |
|---|---|---|
| (Grau et al., 2012) [13] | Deployment support<br>Investment support for manufacturing plants<br>R&D support | FiT, national-market stimulation schemes<br>Grants/cash incentives, reduced-interest loans, public guarantees<br>Support for research programs |
| (Avril et al., 2012) [17] | Market incentives<br><br>Technological and R&D incentives | FiT, subsidies, loans, tax reductions, capacity-driven approaches, TGC<br>R&D subsidies, demonstration programs |
| (Hosenuzzaman et al., 2015) [16] | Market incentives<br>Technological and R&D incentives | FiT, subsidies, loans, TGC<br>R&D funding, demonstration programs |
| (Zhi et al., 2014) [15] | Demand-side policy instruments<br><br><br>Supply-side policy instruments | Feed-in-tariffs, subsidies, net metering, green tags, renewable energy portfolios, financial support, public investment, tax credits, government mandates and regulatory provision<br>Research development and demonstration grants, low-cost loans for manufacturing, tax concessions, subsidized support infrastructure |
| (Deshmukh et al., 2012) [7] | Demand-pull policies<br><br><br>Technology-push policies | FiT, RPS, capital-based incentives or rebates, tax incentives, grants, interest subsidies or low-cost financing, loan guarantees<br>Grants or low-cost loans Tax concessions, R&D grants, training activities, subsidized support infrastructure |
| (Solangi et al., 2011) [5] | No category | FiT, subsidies, incentives, target |
| (Lipp, 2007) [18] | No category | FiT, RPS |
| (P. G. Jordan, 2013) [19] | No category | Tax credits, direct cash financing, property tax incentives, economic development incentives, permitting, loans |

*2.2. Factors Influencing the Deployment of PV Policies in a Global Context*

2.2.1. Factors Influencing the Development of PV Policies Research

There are two main categories of contributing factors to PV policies, namely, the economic and governance status, including GDP and democracy level, and the overall national energy and innovation capabilities of a country, including RE investment and RD expenditure. The first category within the influencing factors relies on the economic performance and governance of a country, which includes GDP (in million USD) and the Democracy Index (DI), as GDP and DI were proven to have an obvious influence on the formation and launch of renewable electricity policies in 18 countries, and a similar conclusion could be drawn, that developed countries generally had more-advanced renewable energy promoting policies compared to most of the developing countries. In 2015, solar power contributed RMB 0.31 trillion to China's GDP growth, which will reach RMB 1.57 trillion in 2030, about 1.1 percent of that year's GDP. In Spain, the solar-PV sector accounts for 0.19% and 0.31% of GDP in direct and in total terms, respectively [20].

However, the current literature fails to find a consistent conclusion as there exist contradictions. The second group of factors relies on fossil-fuel consumption and LCOE (levelized cost of energy) (USD/kWh) and PV penetration, which represent the energy capability of the countries.

### 2.2.2. Influencing Factors Selection

It is found that there exists no research on the comprehensive list of contributing factors affecting solar PV policies. Therefore, we reviewed papers through Science Direct and selected the most-frequently mentioned factors contributing to the difference in solar-PV policy deployment. By reading through the abstracts of all the 200 papers, 16 relevant studies were further reviewed and the factors mentioned were collected in Table 2 below. The seven selected factors contributing to the successful deployment of solar-PV policies are GDP, Democracy index, fossil-fuel consumption, PV penetration, LCOE, RE investment, and R&D expenditure.

**Table 2.** Selection of contributing factors.

| Author | GDP (1) | DI (2) | Fossil (3) | PV (4) | LCOE (5) | RE (6) | R&D (7) |
|---|---|---|---|---|---|---|---|
| (Hess & Mai, 2014) [21] | ✔ (8) | ✔ | ✔ | | | | |
| (Schaffer & Bernauer, 2014) [22] | | ✔ | ✔ | | | | |
| (Morlet & Keirstead, 2013) [23] | ✔ | | | | | | |
| (Rajiv, 2014) [24] | | ✔ | | | | | |
| (Coley & Hess, 2012) [25] | | | | ✔ | | | |
| (Menz & Vachon, 2006) [26] | | ✔ | | | ✔ | | |
| (Lipp, 2007) [18] | | | | ✔ | | ✔ | |
| (Grau et al., 2012) [13] | | | | | ✔ | | |
| (Kwan, 2012) [27] | | | | | ✔ | ✔ | |
| (Ondraczek, Komendantova, & Patt, 2015) [28] | | | | | ✔ | | |
| (Zhang, Zhou, & Zhou, 2014) [29] | | | | | ✔ | ✔ | |
| (Chaianong & Pharino, 2015) [30] | | | | | | ✔ | |
| (Deshmukh et al., 2012) [7] | ✔ | | | | | | ✔ |
| (Hoppmann et al., 2014) [12] | | | | | | | ✔ |
| (Song et al., 2015) [11] | | | | | | | ✔ |
| (Deshmukh et al., 2012) [7] | | | | | | | ✔ |

(1) Gross Domestic Product, million USD. (2) Democracy index, 0–100, an index gathered to reflect how democratic a country is. Related to the willingness and easiness for a country to deploy national supportive policies. (3) Fossil-fuel consumption, %. (4) PV Penetration, %. (5) Levelized cost of electricity, USD/kWh. (6) RE investment, billion USD. (7) R&D expenditure, % of GDP. (8) "✔" indicates the contributing factor is applied in the research.

Moreover, the contributing factors and affected policies were identified in different target countries and the detailed analysis is illustrated in Table 3.

**Table 3.** Literature review on target countries and factors, and affected policies.

| Author and Date | Target Countries | Contributing Factors Studied | Affected Policies | |
|---|---|---|---|---|
| | | | Market Policies | Technology Policies |
| Hess and Mai, 2014 [21] | 18 Asian countries (Japan, South Korea, North Korea, China, Mongolia, Taiwan, Vietnam, Cambodia, Thailand, Malaysia, Singapore, Indonesia, Philippines, Bangladesh, India, Pakistan, Sri Lanka, Myanmar) | GDP<br>DI<br>Fossil | √ (+ve)<br>√ (+ve)<br>√ (−ve) | |
| Schaffer and Bernauer, 2014) [22] | 26 advanced industrialized countries (IEA Member Countries, not specified) | GDP<br>Fossil | √ (+ve)<br>√ (+ve) | |
| Morlet and Keirstead, 2013 [23] | 4 European cities (London, Paris, Berlin, Copenhagen) * Comparative study on governance of energy policies, no applicable result to our study | DI | | |
| Rajiv, 2014 [24] | China and India | PV<br>R&D | √Vina & | √&Dna & |

**Table 3.** *Cont.*

| Author and Date | Target Countries | Contributing Factors Studied | Affected Policies | |
|---|---|---|---|---|
| | | | Market Policies | Technology Policies |
| Coley and Hess, 2012 [25] | US | Fossil | √ossil& | |
| Menz and Vachon, 2006 [26] | US<br>* Result show positive relationship between RPS (market policy) and wind power development | | √Result | |
| Lipp, 2007 [18] | Denmark, Germany, UK<br>* Compared the effectiveness of FiT and RPS to develop RE | | √develo | √develo |
| Grau et al., 2012 [13] | Germany, China | LCOE | | √COEany |
| Kwan, 2012 [27] | US<br>* The affecting factors on solar adoption | | | |
| Ondraczek, Komendantova, and Patt, 2015 [28] | 143 Countries | LCOE | √COECou | |
| Zhang, Zhou, and Zhou, 2014 [29] | China<br>* Proposed an evaluation model to evaluate the balance point for RE investment | | | √evalu |
| Chaianong and Pharino, 2015 [30] | Thailand<br>* Discussed rooftop-PV implementation | | | √Discus |
| Deshmukh et al., 2012 [7] | 7 Countries<br>(Germany, Spain, US, Japan, China, Taiwan, India) | GDP<br>R&D | √ (+ve) | √&Dve |
| Hoppmann et al., 2014 [12] | Germany<br>* Different phases of German FiT policies for PV | | | √pDiffe |
| Song et al., 2015 [11] | China<br>* PV technologies development in China | | | √ PV te |

The text after * supplements the main content or conclusion of the research.

### 2.3. Qualitative Comparative Analysis Methodologies in Energy Policy Research

This paper adopts qualitative comparative analysis (QCA) as the research method. It was first developed by sociologist Charles Ragin in 1987, who was inspired by the binary logic of Boolean algebra. QCA mainly helps to examine the causal relationships between several conditions and an outcome of interest. Initially, it was developed to be used in the field of comparative politics and historical sociology while it is now applied to a much larger context, including management, economics and engineering [31].

There is an increasing amount of research on the application of QCA in the field of energy policy, to link the variable-driven approach and rich qualitative description approach. Never and Betz [32] compared climate policy performance in seven emerging countries and found that a bad ratio of domestic fossil-fuel production to financial power and a environmentally weak civil society are related to weak climate-policy performance using the QCA method. Meanwhile, del Río et al. [33] used the QCA method in their studies to discover that FiT and investment subsidies are supportive towards the repowering of on-shore wind farms. Another case study analyzed the factors contributing to stronger renewable electricity policies in Asia [21]. Therefore, those previous studies of how QCA can be applied to the energy sector affords us the possibility to apply QCA method to solar PV policies.

However, the current scope of the adoption of the methodologies suffers from the limitation that it only focuses on the understanding of the conditions of successful renewable-energy expansion in EU regions, or at the federal-state level. In other words, there is a lack

of research performing a comprehensive review of the global sets of cases, to investigate the combinations of causal conditions from which it is then possible to identify one or more pathways explaining a particular outcome.

## 3. Research Methodology

### 3.1. Selection of Cases

The research chose 20 countries as the leading cases to represent the outcome of policies through using the cumulative installed capacity of solar PV in the year 2014, as shown in Table 4. However, instead of Bulgaria (ranking 20th), South Africa (ranking 21st) was included so that all five continents are fully covered.

**Table 4.** Selection of 20 case countries.

| Countries | Short Names | 2014 Cumulative Installed Capacity/MW | 2014 Ranking |
|---|---|---|---|
| Germany | GE | 38,200 | 1 |
| China | CN | 28,199 | 2 |
| Japan | JP | 23,300 | 3 |
| Italy | IT | 18,460 | 4 |
| US | US | 18,280 | 5 |
| France | FR | 5660 | 6 |
| Spain | SP | 5358 | 7 |
| UK | UK | 5104 | 8 |
| Australia | AU | 4136 | 9 |
| Belgium | BE | 3074 | 10 |
| India | IN | 2936 | 11 |
| Greece | GR | 2595 | 12 |
| Korea | KR | 2384 | 13 |
| Czech Republic | CZ | 2134 | 14 |
| Canada | CA | 1710 | 15 |
| Thailand | TH | 1299 | 16 |
| Romania | RM | 1219 | 17 |
| Netherlands | GE | 1123 | 18 |
| Switzerland | CN | 1076 | 19 |
| South Africa | JP | 922 | 21 |

Sources: statistical review of world energy June 2015 (BP, 2015) [34]. The data from Renewable energy 2014 Cumulative Installed Capacity in BP Statistical Review Of World Energy June 2015 is processed and consolidated into this table.

### 3.2. Policy Outcomes

- Market-based policies index

After a ful review of the literature, three sets of data were selected to represent the outcome of the market-based policies, which includes Target-to-capacity ratio, FiT, and Others (which includes subsidies, tax, loans, and TGC/RPS).

The calibrated data (shown in brackets in Table 5) of the above three indicators is then weighted evenly to produce the market-based policies index.

- Technological-R&D-based policies index

The research considered two sets of policies to evaluate the Technological-R&D policies in terms of PV-R&D funding and solar-demonstration programs.

PV-R&D funding: the data was obtained from the IEA trend report 2014, while the cases of South Africa and India were estimated by assuming that 50% of their renewable-energy R&D funding was invested in solar PV energy (USD 0.003 bn and USD 0.3 bn, respectively), deriving the number of USD 1.5 million and USD 150 million.

Demonstration programs: since the number and magnitude of demonstration programs in various countries varies significantly, a three-scale score (0, 0.5, 1) was allocated for

this factor. While 1 represents "present at national level", 0.5 represents "present at regional level" and 0 represents "absent," regarding demonstration programs in the country.

The calibrated data (shown in brackets in Table 6 below) of the above two indicators was then weighted evenly to produce the Technological and R&D Incentive Index score.

**Table 5.** Summary of indicators used in market-based policies index.

| Countries | Target-to-Capacity Ratio2012/2020 | FiT (1) USD/KWh Below 10 kW | Others (Subsidies, Tax, Loans, and TGC/RPS) | Market-Based Policies Index |
|---|---|---|---|---|
| Germany | 0.66 | 0.19 (0.23) | 3 (0.75) | 0.55 |
| China | 1.00 | 0.15 (0.18) | 4 (1.00) | 0.73 |
| Japan | 0.89 | 0.32 (0.39) | 4 (1.00) | 0.76 |
| Italy | 1.00 | 0 (0) | 2 (0.50) | 0.50 |
| US | 0.60 | 0.20 (0.24) | 4 (1.00) | 0.61 |
| France | 0.56 | 0.42 (0.51) | 4 (1.00) | 0.69 |
| Spain | 0.72 | 0 (0) | 2 (0.50) | 0.41 |
| UK | 0.37 | 0.18 (0.22) | 4 (1.00) | 0.53 |
| Australia | 0.50 | 0.60 (0.73) | 2 (0.50) | 0.58 |
| Belgium | 0.98 | 0.33 (0.40) | 4 (1.00) | 0.79 |
| India | 0.80 | 0.12 (0.15) | 4 (1.00) | 0.65 |
| Greece | 0.93 | 0.48 (0.59) | 2 (0.50) | 0.67 |
| Korea | 0.64 | 0 (0) | 3 (0.75) | 0.46 |
| Czech Republic | 0.72 | 0.65 (0.79) | 2 (0.50) | 0.67 |
| Canada | 0.80 | 0.82 (1) | 3 (0.75) | 0.85 |
| Thailand | 0.77 | 0.20 (0.24) | 2 (0.50) | 0.50 |
| Romania | 0.95 | 0.38 (0.46) | 2 (0.50) | 0.64 |
| Netherlands | 0.87 | 0.52 (0.63) | 3 (0.75) | 0.75 |
| Switzerland | 0.79 | 0.28 (0.34) | 2 (0.50) | 0.54 |
| South Africa | 0.08 | 0.20 (0.24) | 4 (1.00) | 0.44 |

Source: (1) Data for FiT in most countries from (PV-Tech, 2016) [35], data for KR from (IEA, 2012a) [36]. The data from the Country FiT statistics in the http://www.pv-tech.org/tariff_watch/list (accessed on 29 March 2022) and the ANNEX B: Energy balances and key statistical data in Energy Policies of IEA Countries 2012 Review The Republic of Korea is processed and consolidated into this table.

**Table 6.** Summary of indicators used in technological R&D policies index.

| Countries | PV R&D Funding in 2013 (1)/M USD | Demonstration Programs (2) | Technological R&D Based Policies Index |
|---|---|---|---|
| Germany | 250.60 (1) | 1 | 1.00 |
| China | 79.00 (0.32) | 1 | 0.66 |
| Japan | 89.80 (0.36) | 1 | 0.68 |
| Italy | 7.70 (0.03) | 0.5 | 0.27 |
| US | 194.40 (0.78) | 0.5 | 0.64 |
| France | 5.30 (0.02) | 0 | 0.01 |
| Spain | 23.90 (0.10) | 0 | 0.05 |
| UK | 70.80 (0.28) | 0.5 | 0.39 |
| Australia | 170.20 (0.68) | 1 | 0.84 |
| Belgium | 4.20 (0.02) | 1 | 0.51 |
| India | 150.00 (0.60) | 0.5 | 0.55 |
| Greece | 35.00 (0.14) | 0 | 0.07 |
| Korea | 202.40 (0.81) | 0 | 0.40 |
| Czech Republic | 99.40 (0.40) | 0 | 0.20 |
| Canada | 11.70 (0.05) | 1 | 0.52 |
| Thailand | 28.00 (0.11) | 0.5 | 0.31 |
| Romania | 0.46 (0) | 0 | 0.00 |
| Netherlands | 35.00 (0.14) | 0 | 0.07 |
| Switzerland | 0.74 | 1 | 0.50 |
| South Africa | 1.50 | 0 | 0.00 |

Sources (1) Data for PV R&D Funding were obtained from (IEA, 2014c) [37] for most of the countries, besides TH: (IEA, 2014b) [38]; SW: (Husser, GmbH, and Aarau, 2015) [39]; GR: (IEA, 2012b) [38]; RM: (Sandu and Dinges, 2007) [40] (0.46 = 4.19 × 33% × 33%, assume 33% of RES, and 33% of solar (Teodoreanu, 2013) [41]) (CZ: 0.74 = 1.19 × 31% × 27%)(Gerden, 2015) [42] (2) Data for demonstration programs obtained from (IEA, 2015b) [43].

### 3.3. Contributing Factors Calibration

Seven factors have been selected to represent the most relevant factors to the policy outputs. However, as suggested by Schneider and Wagemann [44], the greater the number of factors is, the more possible combinations will exist, and a much higher number of cases would be required for a more meaningful outcome. Given the fact that we are limited to 20 cases, only four factors were set to be chosen to evaluate the different combinations of factors, which amounts to a total of 16 (=24) combinations.

In order to select four factors out of the seven factors, a survey was designed and sent out to experts in the field of solar-PV energy. The survey was conducted through an online platform, and 20 experts from different countries were chosen to fill in the survey, which covered most parts of the countries listed in the research. In the survey, experts were asked to rate the importance of each factor in two sections scale: (1) the importance of factors influencing market-based Solar-PV policies, and (2) the importance of factors influencing Technological-R&D-based policies. Table 7, below, shows the overall rating for each factor based upon the 13 pieces of feedback from the experts, where 1 represents "not at all important" and 5 represents "absolutely critical". From the results, it is obvious that fossil-fuel consumption and LCOE could be considered as the contributing factors for market-based policies; with RE investment and R&D Expenditure possibly contributing more to Technological-R&D-based policies.

**Table 7.** Summary of indicators used in technological R&D policies index.

| Contributing Factors | Market-Based Policies (M) | Technological-R&D-Based Policies (T) | Contribute to Policy | Final Decision |
|---|---|---|---|---|
| GDP | 3 | 3.5 | ~ | M&T |
| Democracy Index | 2.4 | 2.6 | ~ | M |
| Fossil-Fuel Consumption | 4.1 | 3.6 | M | M |
| LCOE | 4.7 | 3.8 | M | M |
| PV Penetration | 3.6 | 3.5 | ~ | T |
| RE Investment | 3.9 | 4.1 | T | T |
| R&D Expenditure | 3.7 | 4.5 | T | T |

For GDP and Democracy index, we decided to include both as indicators for market-based policies according to research carried out by Hess and Mai [21], who found out that a higher GDP and higher democracy index are related to relatively advanced renewable electricity policies that belong to market-based policies by our definition. We also decided to consider GDP as a factor for Technological-R&D-based policies, since there has been research that identified the relationship between GDP and energy policies as a whole [23]. Lastly, as it can be seen that the value of PV penetration for market-based and Technological-R&D-based policies are close (3.6 and 3.5, respectively), we then selected it as a contributing factor for Technological-R&D-based policies according to our research, which demonstrates a clear relationship between the installed capacity (PV penetration) and Technological-R&D-based policies. Based upon experts' advice and reference to research papers, four factors for both market-based and Technological-R&D-based policies were decided.

Raw data was calibrated and standardized to dummy variables (0–1) for conducting fsQCA analysis in the next step. For the purpose of standardization, all calibrations follow the formula as shown below:

$$\text{Calibrated data} = \frac{(X - W_{\min})}{(W_{\max} - W_{\min})}$$

where X represents the data of a given country, while $W_{\min}$ and $W_{\max}$ represent the world minimum and maximum for that particular set of data. World maximum and minimum were used to calibrate data so that countries not included in the list could also make use of

the QCA results obtained from this research. Raw data and the calibrated data is shown in Table 8 below.

**Table 8.** Raw data for contributing factors.

| Country | GDP (1) | DI (2) | Fossil (3) | PV (4) | LCOE (5) | RE (6) | R&D (7) |
|---|---|---|---|---|---|---|---|
| Germany | 3,868,291 (0.22) | 8.64 (0.85) | 81.25 (0.72) | 7.1 (0.90) | 0.41 (1.00) | 10.1 (0.19) | 2.85 (0.69) |
| China | 10,354,832 (0.59) | 3 (0.22) | 89.15 (0.84) | 0.7 (0.09) | 0.09 (0.22) | 54.2 (1.00) | 2.08 (0.50) |
| Japan | 4,601,461 (0.26) | 8.08 (0.79) | 93.07 (0.90) | 2.57 (0.32) | 0.29 (0.71) | 28.6 (0.53) | 3.47 (0.84) |
| Italy | 2,141,161 (0.12) | 7.85 (0.76) | 81.4 (0.72) | 7.95 (1.00) | 0.24 (0.59) | 3.6 (0.07) | 1.26 (0.30) |
| US | 17,419,000 (1.00) | 8.11 (0.79) | 86.3 (0.79) | 0.65 (0.08) | 0.38 (0.93) | 36.7 (0.68) | 2.73 (0.66) |
| France | 2,829,192 (0.16) | 8.04 (0.79) | 49.64 (0.24) | 1.38 (0.17) | 0.2 (0.49) | 2.9 (0.05) | 2.23 (0.54) |
| Spain | 1,381,342 (0.08) | 8.05 (0.79) | 71.58 (0.57) | 3.46 (0.44) | 0.25 (0.61) | 0.4 (0.01) | 1.24 (0.30) |
| UK | 2,988,893 (0.17) | 8.31 (0.82) | 84.51 (0.77) | 1.55 (0.20) | 0.26 (0.63) | 12.4 (0.23) | 1.63 (0.39) |
| Australia | 1,454,675 (0.08) | 9.01 (0.90) | 68.62 (0.53) | 2.37 (0.30) | 0.27 (0.66) | 4.4 (0.08) | 2.13 (0.51) |
| Belgium | 531,547 (0.03) | 7.93 (0.77) | 81.63 (0.72) | 3.6 (0.45) | 0.3 (0.73) | 1.6 (0.03) | 2.28 (0.55) |
| India | 2,048,517 (0.12) | 7.92 (0.77) | 91.96 (0.88) | 0.65 (0.08) | 0.18 (0.44) | 6 (0.11) | 0.81 (0.20) |
| Greece | 235,547 (0.01) | 7.45 (0.72) | 88.89 (0.83) | 7.6 (0.96) | 0.11 (0.27) | 1.94 (0.04) | 0.8 (0.19) |
| Korea | 1,410,383 (0.08) | 8.06 (0.79) | 86.31 (0.79) | 0.6 (0.08) | 0.07 (0.17) | 1 (0.02) | 4.15 (1.00) |
| Czech Republic | 205,270 (0.01) | 7.94 (0.78) | 78.24 (0.67) | 3.8 (0.48) | 0.24 (0.59) | 1.6 (0.03) | 1.92 (0.46) |
| Canada | 1,785,387 (0.10) | 9.08 (0.90) | 65.52 (0.48) | 0.4 (0.05) | 0.24 (0.59) | 6.5 (0.12) | 1.62 (0.39) |
| Thailand | 404,824 (0.02) | 5.39 (0.49) | 97.78 (0.97) | 1.1 (0.14) | 0.14 (0.34) | 1.5 (0.03) | 0.24 (0.06) |
| Romania | 199,044 (0.01) | 6.68 (0.63) | 75.37 (0.63) | 2.7 (0.34) | 0.24 (0.59) | 1.15 (0.02) | 0.39 (0.09) |
| Netherlands | 879,319 (0.05) | 8.92 (0.89) | 95.56 (0.93) | 1 (0.13) | 0.02 (0.05) | 6.7 (0.12) | 1.98 (0.48) |
| Switzerland | 701,037 (0.04) | 9.09 (0.91) | 46.69 (0.20) | 1.8 (0.23) | 0.24 (0.59) | 5.8 (0.11) | 2.96 (0.71) |
| South Africa | 350,085 (0.02) | 7.82 (0.76) | 96.45 (0.95) | 0.67 (0.08) | 0.16 (0.39) | 4.9 (0.09) | 0.73 (0.18) |

Source: data resources are referred as "superscript" in the above table. (1) GDP (million USD): World Bank 2014 GDP; (2) Democracy Index 2014: [45]; (3) Fossil-fuel consumption in 2014 (%): [34]; (4) PV Penetration (%): [44]; (5) LCOE (USD/kWh): [46] except: KR: [47], GR/NT: [48], TH: [49], CZ/CA/SW/RM: [50]; (6) RE investment (billion USD): [51] except: BE: [52], KR: [53], RM: [54], CZ: [41], SW: [55]; (7) R&D expenditure as % of GDP (%): [56] except TH: [57] QCA analysis Con.

## 4. QCA Results and Discussion

### 4.1. QCA Outcome 1: Market-Based Policies

The four factors considered for market-based policies are: GDP, Democracy index, Fossil-fuel consumption and LCOE. GDP, Democracy and Fossil fuel were chosen based on a study that used QCA to find out which factors affect renewable electricity policy (categorized under market-based policies in this paper) in 18 Asian countries [21]. The results are shown in Table 9. It was found that GDP, democracy, and fossil-fuel reserves indeed contribute to a better policy that leads to higher renewable electricity generation. LCOE was chosen as a contributing factor to market-based policies due to its close relationship with the solar-PV market. As a result, the following table indicates the combination of factors that influence market-based policies, target, FiT and other policies, respectively.

The QCA results for market-based policies, Target, and Others turn out to be very similar, with two significant combinations which are read as follows:

The first significant combination (LCOE*fossil*democracy) that leads to stronger market-based policies consist of countries with a high LCOE level, high fossil-fuel ratio, and high democracy index, for example, GE, IT, US, JP, SP, UK, AU, BE, CZ, and RM. This outcome is supported by several past studies. For instance, the research conducted by Schaffer and Bernauer [23] shows that the heavier is a reliance on fossil fuel consumption, the better are market-based policies. Moreover, the research completed by Hess and Mai [22] suggested that the democracy level of a country has a positive relationship with the available PV-related market policies.

In terms of the second significant combination (~LCOE*Fossil*~Democracy) presented in the QCA results, developing countries such as China and Thailand tend to have better market policies. A potential reason for this could be that their low LCOE brings them a high fossil fuel %, which then indicates a low Democracy index and helps the facilitation

of market policy from central government. This result shares some similarities with our review of the literature.

However, it does not necessarily mean that high fossil-fuel consumption and a low Democracy index will definitely result in better market-based solar energy policies. For instance, in China's 1980~1985 period, when authoritarian rule was dominant and fossil remained the key source of energy. No specific, important solar policy was implemented by the Chinese government. The reason for this phenomenon is that the cost of PV solar regeneration was too high [16].

It is noticeable that, for all the above-mentioned models and paths, GDP was absent, despite being considered among the contributing factors. This means that GDP is not a significant contributing factor towards stronger market-based policies. One possible reason for this could be that there exist different motivations for countries with varied GDPs to invest in RE, as explained by our interview with solar energy analysts who replied: "Countries with high gross GDP could be motivated to invest in RE in order to decarbonize the energy mix and because they have the money. However, countries with low GDP may need RE in order to provide access to electricity (especially in rural regions). Therefore, the importance of investment in RE may not be easy to correlate with GDP because the country's motivation for RE changes as its GDP grows."

**Table 9.** QCA results for market-based policies.

| Policy Outcomes | Combination of Contributing Factors | Coverage (2) | Consistency (3) | Countries |
|---|---|---|---|---|
| Market-based policies | LCOE*Fossil *Democracy (4) | 0.686102 | 0.929654 | GE, IT, US, JP, SP, UK, AU, BE, CZ, RM |
| | ~LCOE*Fossil *~Democracy (5) | 0.364217 | 0.993464 | CN, TH |
| Target | LCOE*fossil *Democracy | 0.607887 | 0.884199 | GE, IT, US, JP, SP, UK, AU, BE, CZ, RM |
| | ~LCOE*Fossil *~Democracy | 0.329613 | 0.965142 | CN, TH |
| Others (Subsidies, tax, loans, TGC/RPS) | LCOE*Fossil *Democracy | 0.591333 | 0.959957 | GE, IT, US, JP, SP, UK, AU, BE, CZ, RM |
| | ~LCOE*Fossil *~Democracy | 0.30533 | 0.997821 | CN, TH |
| FiT (1) | LCOE*Fossil *Democracy | 0.651648 | 0.641775 | Not applicable |

(1) For FiT, there is no path with a consistency higher than 0.85; therefore, no result was shown, which means that no combination of factors is sufficient to lead to the policy outcome. (2) Coverage refers to the number of cases with the outcome that is represented by a specific combination of contributing factors [58]. A simple mathematical representation for the calculation of the coverage score can be written as:

$$Coverage = \frac{\sum \min(X,Y)}{\sum Y}$$

where X is the membership score of a contributing factor and Y is the membership score of the outcome. (3) Consistency is used to determine the extent to which a combination of factors leads to the outcome [58]. The calculation of the consistency score is similar to that of consistency, which can be simply represented as:

$$Consistency \ (X < Y) = \frac{\sum \min(X,Y)}{\sum X}$$

(4) "*" is a linking symbol between contributing factors, indicating the intersection relation of "and", that is, the connected contributing factors need to be satisfied at the same time. (5) "~" is used to indicate the logical relation of "not", i.e. the contributing factor does not exist.

### 4.2. QCA Outcome 2: Technological-R&D-Based Policies

For R&D incentives, factors considered were: GDP, RE investment, R&D funding and PV penetration. The QCA analysis results are shown in Table 10. A single combination was generated for all three models with satisfactory consistency and coverage level. The path to stronger Technological-R&D-based policies and stronger demonstration projects

includes countries with high R&D expenditure as % of GDP, high RE investment and low PV penetration level. Countries that fit into this path include CN, JP and US. The path to stronger R&D funding is similar to the above-mentioned combination, including countries with high R&D expenditure as % of GDP, high RE investment, low PV penetration level, and high GDP. This shows that the presence of a high GDP could be a sign for countries to invest more in solar R&D research. Countries that fit into this path are CN and the US.

**Table 10.** QCA results for technological-R&D-based policies.

| Policy Outcomes | Combination of Contributing Factors | Coverage | Consistency | Countries |
|---|---|---|---|---|
| Technological-R&D-based policies | R&D*RE invest *~PV penetration | 0.346806 | 0.923611 | CN, JP, US |
| R&D funding | R&D*RE invest *~PV penetration *GDP | 0.327055 | 0.848889 | CN, US |
| Demonstration programs | R&D*RE invest *~PV penetration | 0.246316 | 0.812500 | CN, JP, US |

As calibrated by the QCA method, the resulting path that leads to more comprehensive Technological-R&D-based policies, including R&D funding and demonstration projects, consists of countries where the R&D expenditure as % of GDP is high, RE investment is high, but the PV penetration level is relatively low. Countries that are most suitable to represent this path are China, Japan and US. Correspondingly, many researchers shared similar views on this subject. For instance, Deshmukh et al. [8] discovered that national macroeconomic conditions have a significant impact on solar support policies, which would subsequently affect the R&D-related funding for solar. Similarly, the accessibility of capital for R&D funding and the penetration rate of large-scale photovoltaic (LS-PV) projects played essential roles in the rapid PV development of China [26].

## 5. Overall Finding and Policy Implications

### 5.1. General Policy Implications within the Countries

To simplify the finding, especially for market-based policies, there are two combinations of factors that could lead to successful policy outcomes, as represented in Table 11 below, where H represents a high level and L represents a low level. This overall finding can be read as: high/low GDP with high democracy level, high fossil-fuel consumption and high LCOE are related to the deployment of market-based policies, which include target, FiT and others (subsidies, tax, loans, TGC/RPS); while high/low GDP, a low level of PV penetration, high RE investment, and high R&D expenditure contribute to more successful Technological-R&D-based policies, which include R&D funding and demonstration programs.

**Table 11.** Summary of findings from QCA results.

| Policy Outcomes | GDP (1) | DI (2) | Fossil (3) | PV (4) | LCOE (5) | RE (6) | R&D (7) |
|---|---|---|---|---|---|---|---|
| Market-based policies | H/L | H | H | H | - | - | - |
| Target | H/L | H | H | H | - | - | - |
| FiT | N.a. | N.a. | N.a. | N.a. | - | - | - |
| Others | H/L | H | H | H | - | - | - |
| Technological-R&D-based policies | H/L | - | - | - | L | H | H |
| R&D funding | H | - | - | - | L | H | H |
| Demonstration program | H/L | - | - | - | L | H | H |

(1) Gross Domestic Product, million USD. (2) Democracy index, 0–100, an index gathered to reflect how democratic a country is. Related to the willingness and easiness for a country to deploy national supportive policies. (3) Fossil-fuel consumption, %. (4) PV Penetration, %. (5) Levelized cost of electricity, USD/kWh. (6) RE investment, billion USD. (7) R&D expenditure, % of GDP.

This finding can, therefore, be applied during the policymaking stage of a country that aims to develop its national solar-PV market. For example, a country that is democratic and has high fossil-fuel consumption is advised to adopt market-based policies; while a country with a low PV penetration level, high RE investment and high R&D expenditure could achieve more a successful outcome with Technological-R&D-based policies. However, we should acknowledge that it is almost impossible to find the best policy combinations, just as stated by Ekins "No optimal model has emerged, and probably none will do so in the contexts that is shaped by different histories and cultures". Therefore, the findings of this research aim to serve only as guidance for countries during the policymaking stage.

The current study identified that more-democratic countries prefer to rely on coal, natural gas, and modern renewable sources (nuclear, biomass, wind, and others), but less-democratic countries tend to become more dependent on oil and natural gas for their own development.

### 5.2. Mechanism of the Policies and Strategies

Germany has always been the leading country in the solar PV market. With a high level of democracy index, fossil-fuel consumption and LCOE, Germany is a successful case of the deployment of market-based policies. Firstly, several points could be highlighted: Germany implemented the first feed-in-tariff (FIT) incentive in the world in 1991, which ensures a guaranteed price to be paid by electricity utility providers to independent producers for energy distributed into the grid, providing long-term financial stability to investors [6]. In 2004, the government released the EEG Amendment Act, stating that the on-grid price was EUR 0.547–0.624/kWh, which would be reduced 5% each year for the following 20 years. The government then amended the ACT in 2008, reducing the FiT by 15%, and again reduced the FiT in 2010 and 2011. Secondly, Germany has a target to achieve 35% of the electrical mix as generated by renewables by 2020 and 80% by 2050, according to the Renewable Portfolio Standards [6]. Thirdly, Germany's Federal Ministry of Environment provides direct subsidies and tax credits for PV products and equipment, as well as for various solar projects. For example, investment in PV products and equipment production is deductible for the 12.5–27.5% investment yield tax [15].

In contrast, China mainly promotes the development of the solar PV by increasing R&D investment, and seldom uses market-based policies. According to the IEA's report in 2013, the annual investment in R&D and demonstrations for solar power was USD 80 million, which has successfully led to lowered system prices and enhanced the efficiencies of different kinds of technologies. Moreover, China initiated many demonstration projects in the past; the more recent ones are "Large-scale PV Power Station Concession Bidding", which involved the installation of 4.3 GW large-scale solar power stations between 2009 and 2012, while the "Golden-Sun Pilot Project," which supported 700 different solar-PV-generation projects, was also initiated in 2009 [15].

It is worth noting that policies are not static. A country will adjust its policy direction according to its own national conditions at different stages. The US was strong in both of its market-based policies as well as Technological-R&D-based policies. Budgets for R&D and demonstration programs are mainly from Federal budgets. In 2014, a total of USD 439 million went into R&D while USD 26 million went into demonstration incentives. Other recent R&D incentives include the SunShot Initiative that was launched by the Secretary of Energy in February 2011. This program aims to encourage solar-technology innovations to bring down solar systems' competitive costs. As a part of this program, private companies and laboratories were financially supported to reduce the cost of utility-scale solar electricity to USD 0.06 per kW/h [59]. With the characteristics of a high democracy index, high fossil-fuel consumption and high LCOE, the US is also a successful case of the adoption of market-based polices. FiT was implemented at state- and city-government levels in 2008. In 2014, six states and 17 utilities were offering FiT at different rates, with an effective period of 20 years in general [60]. Tax credits are granted by the federal US government, which is the main solar-PV supportive policy in the US. It offers a 30% investment tax credit for

commercial-grid-connected PV systems and a 30% tax credit for residential-grid-connected PV systems, with an annual cap of USD 2000 per system [17]. Solar was first involved in the US RPS in 2007, and is currently implemented in 29 states.

Although QCA results show that Japan appears in both market-based and technological-R&D-based configurations at the same time, this does not mean that Japan can implement two policies at the same time as the United States, as Japan is far less-democratized than the United States. The Japanese government focused on R&D incentives and demonstration programs in the early stage, which allowed the country to improve related technologies. It started as early as 1973, when the government proposed the "Development Plan of Renewable Energy Technologies", which was then followed by "Development Plan of Energy Saving" (1978) and the "Development Plan of Environment Protection" (1989) to promote PV R&D. Low-interest loans and taxes were also implemented at the same time, to encourage companies' investment in PV R&D [14].

## 6. Conclusions and Implications

There is a large body of policy literature on solar-PV policies, and most of them consider one particular country's solar-PV policy rather than constituting global comparative studies. This research conducted a literature review on global solar-PV policies and implemented a useful tool in the area of policy studies, the QCA method. Various policies that have been deployed by countries were categorized into Market-based and Technological-R&D-based policies and seven possible contributing factors that could lead to sustainable policy deployment were identified [6]. Data input for both policy outcomes and contributing factors was collected and calibrated using twenty leading countries as case countries. The QCA method was then applied to discover a causal relationship between contributing factors and the policy outcomes. The research provides a QCA framework to make a comparative study at a cross-border scale.

The main finding of this research is the discovery of the causal relationship between the selected contributing factors and sustainability of the policy outcomes, which can be put simply as high/low GDP with a high democracy level, high fossil-fuel consumption and high LCOE as being related to the deployment of market-based policies, which includes target, FiT and others (subsidies, tax, loans, TGC/RPS); while high/low GDP, low level of PV penetration, high RE investment, and high R&D expenditure contributes to more-successful Technological-R&D-based policies, which include R&D funding and demonstration programs. Compared with other Asian countries, the advantages of China's policies and strategies in solar R&D research lie in the setting of special funds for the construction of new energy bases, which can significantly strengthen the allocation of factors such as land, energy, and capital, and effectively guarantee the production-factor needs of new energy companies and the construction-factor needs of new energy-industry projects [60]. In addition, the importation of turnkey solutions, economies of scale, and lower production costs contribute to China's domination over the production of solar PV [61].

The main contribution of this research is the development of a QCA model framework that discovered the causal relationship between the identified contributing factors and two main categories of policy outcomes. Furthermore, this framework is not limited to the twenty case countries that were considered in this research, but can be easily applied to any other country (or countries) in the world, hence offering advice for countries that are considering the sustainable deployment of solar PV policies during their policymaking stage. Despite this research only selecting twenty case countries and seven contributing factors that might not be exclusive, the established framework remains useful and replicable. Moreover, though QCA is proven to be a useful tool to understand and analyze the complex causality of the contributing factors, its conclusions are very context-sensitive, depending heavily on the selected cases and contributing factors used in the study. It is therefore possible for future research to expand the number of case countries to obtain more convincing QCA results, as well as to test the effectiveness of other contributing factors that might be relevant.

**Author Contributions:** Conceptualization, F.Y.; Formal analysis, Y.F.; Methodology, Y.F. and Y.W.; Resources, Y.L.; Writing—original draft, F.Y. and Y.W.; Writing—review & editing, S.Y. All authors have read and agreed to the published version of the manuscript.

**Funding:** This project was supported by the National Natural Science Foundation of China (No. 52078374) and China Postdoctoral Science Foundation (Grant Number 2020M680442 and 2021T140061).

**Informed Consent Statement:** Not applicable.

**Data Availability Statement:** Data is contained within the article.

**Conflicts of Interest:** The authors declare no conflict of interest.

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
