# Peer review of "Pathways to Sustainable Deployment of Solar Photovoltaic Policies in 20 Leading Countries Using a Qualitative Comparative Analysis"

_sustainability, doi:10.3390/su14105858_

Round 1

Reviewer 1 Report

The authors study the combinations of the factors that support Solar PV deployment in 20 leading countries using a Qualitative Comparative Analysis (QCA) method. The 20 leading countries are chosen from the leading 2014 cumulative installed capacity, except by substitution of Bulgaria with South Africa to showcase all five continents. Based on the QCA results, the authors conclude that high democracy level countries with high fossil fuel consumption and high LCOE tend to use market-based policies such as target, FiT, subsidies, etc. On the other hand, countries with a low level of PV penetration, high RE investment, and high R&D expenditure are related to successful technological R&D-based policies such as R&D funding and demonstration programs. In general, the article has described this relationship adequately with the QCA method.

Some minor revisions might be required:

  1. The authors mention that “In order to select four factors out of the seven factors, a survey was designed and sent out to experts in the field of solar PV energy.”. Please clarify the number of experts, how they are chosen, and nationality (are they from the 20 countries?).
  2. The authors need to explain how to process the raw data in Table 8 to select the combination of contributing factors, and how to calculate coverage and consistency needs to be briefly explained. Formulas/equations can be used in the article to make it self-contained.

Reviewer 2 Report

The article investigates the pathways and combinations of the factors of the sustainable development of Solar Photovoltaic policies using a QCA analysis of 20 leading countries. The main finding of this research is the causal relationship between selected contributing factors and sustainability of the policy outcomes. High/low GDP with high democracy level, high fossil fuel consumption and high LCOE is related to the deployment of market based policies which includes target, FiT and others (subsidies, tax, loans, TGC/RPS); while high/low GDP, low level of PV penetration, high RE investment, and high R&D expenditure contributes to more successful Technological R&D based policies which include R&D funding and demonstration programs. In a word, this paper affords a global scope on the development of solar photovoltaic policies.

  1. How about the contribution of solar photovoltaic industry to GDP in 20 leading countries? This may be closely related to the solar photovoltaic policies.
  2. Please add the full title or related explanations of the abbreviations in the paper.
  3. Please correct the grammatical mistakes in the article.

Reviewer 3 Report

This manuscript investigated the pathways and combinations of the factors of the sustainable development of Solar Photovoltaic policies using a qualitative comparative analysis (QCA) analysis of 20 leading countries, to identify the causal relationship between a combination of contributing factors and the various solar PV policy outcomes towards sustainability in different contexts globally through the QCA method.  It tried to answer which contributing factor (combination of factors) is related to sustainable solar PV market development through supportive policies.

Before the publication in Sustainability, some issues should be addressed.

  1. In the manuscript, the authors' name are written incorrectly. Line 5, “Yujie Lu 1 , Fangxin Yi 2,*, Yangtian Feng 3 , Shaocong yu 4and Wang Yujuan”

  1. Line 16, “The paper investigates the pathways and combinations of the factors of the sustainable development of Solar Photovoltaic policies”. “the factors of” should be “the factors for”

  1. Line 254, “high fossil fuel %”. This writing may be incorrect, may be it can be written as “high fossil fuel ratio”?

  1. Line 390, “time to encourage companies’ investment in PV R&D. [12].” Please check out the punctuation.

  1. Compared with other Asian countries, what are the advantages of China's policies and strategies in solar R&D researches?
